# The Impact of Sustainable Management Strategies of Sports Apparel Brands on Brand Reliability and Purchase Intention through Single Person Media during COVID-19 Pandemic: A Path Analysis

Taerin Chung [1,*] , Kwang-Yong Lee [2,*] and Uk Kim [1]

1   Department of Life Sports, Dankook University, Cheonan 31116, Korea; uk05180@gmail.com
2   Department of Sports Science, Kyonggi University, Suwon 16227, Korea
*   Correspondence: raytonchung@gmail.com (T.C.); guifarro@naver.com (K.-Y.L.); Tel.: +82-10-6886-8303 (T.C.)

**Abstract:** Recently, a variety of efforts have been taken to convey sustainable management strategies of sports apparel brands through single-person media. However, there is a lack of theoretical information on the path that leads these corporate marketing activities to brand reliability and purchase intention of consumers. Therefore, this study aims to analyze the path through which the sustainable management strategy established by a sports apparel brand affects the brand awareness and reliability, as well as the purchase intention of consumers when experiencing this strategy through single-person media. The results are summarized as follows: Firstly, the sustainable management strategies of a sports apparel brand carried out through single-person media had statistically significant positive impacts on the benevolence reliability perceived by single-person media viewers. Secondly, benevolence also had a statistically significant positive impact on the consumers' purchase intention. It is expected that the results of this study will serve as an important resource for the methods of utilizing sustainable management strategies among sports apparel brands in the future.

**Keywords:** path analysis; sustainable management strategy; sports apparel brands; brand reliability; purchase intention; single person media

## 1. Introduction

### 1.1. Background

The establishment of sustainable management strategies has recently become an essential element among sports apparel brands [1,2]. A sustainable management strategy conveys to consumers that the brand contributes to sustainable social development, which consists of the economic, social, and environmental responsibilities that the brand has. Economic responsibility refers to the contribution of corporate profitability to the sustainability of the economy to which the brand belongs [3,4]. Social responsibility refers to taking an interest in and action on social issues such as disadvantaged groups and wealth distribution bias. Environmental responsibility refers to demonstrating a commitment to environmental conservation through actions such as resource-saving and recycling. Sports apparel brands that use such sustainable management strategies as their main marketing strategy include Patagonia and Freitag, who have successfully developed high levels of consumer consumption [3–5]. The use of sustainable management strategies indicates both a brand's capability and its proper use of resources and has become a new way of judging sports apparel brands among consumers via criteria such as new apparel fabric development or resource recycling and upcycling [1,6].

As the marketing strategies of brands have become diversified from the traditional media (such as television, radio, and newspapers) and become more personalized in recent years, social networking services (SNS)-based single person media have gained much

popularity among sports apparel brands and are expected to become the new means of distributing information related to corporate sustainable strategies [7,8]. Moreover, the individual isolation caused by the COVID-19 pandemic due to lockdowns, social distancing, and quarantine has also contributed to the rapid growth of single-person media [9–11]. Considering these changes in the apparel market, advertising sustainable management strategies using single-person media may be a highly effective method for sports apparel brands. Considering that single-person media content is usually watched on smartphone, which are very portable, it can exert massive amounts of social influence with high publicity from famous figures [8,12]. A show host and multiple viewers have real-time text or live call conversations in unrefined expression with a personal computer (including laptop) or smartphone as its method. It is a unique precision that existing media does not have, and at the same time, it can be an advantage [13–15].

Recently, efforts have been made in various ways to communicate the sustainable management strategies of sports apparel brands using single-person media. However, there is a lack of theoretical understanding of the path through which these corporate marketing activities affect the brand reliability and purchase intention of consumers, and therefore, an academic basis that can demonstrate the marketing effects of sustainable management is needed.

### 1.2. Purpose of Research

The purpose of this study is to analyze the path through which sustainable management strategies established by sports apparel brands affect brand reliability and purchase intention that consumers perceive when these strategies are conveyed via single-person media.

## 2. Theoretical Backgrounds and Hypothesis

### 2.1. Sustainable Strategies of Sports Apparel Companies Conveyed through Single Person Media and Brand Reliability

For a company that establishes and executes sustainable strategies, brand reliability—the trust that consumers have in the brand—is a key factor. This trust, the possession of unconscious beliefs about the brand's reliability based on the messages and values surrounding economic, social, and environmental responsibilities that the brand conveys as defined by [16] as a psychological state with the intention to tolerate vulnerability based on the positive expectation about the intention or behavior of other people. Ref. [17] defined it as the expectation that the transaction counterpart will want to cooperate and fulfill its obligations and responsibilities based on the belief that the counterpart's words and promises can be trusted, and that it is committed to fulfilling its obligations in the exchange relationship. On the other hand, Ref. [18] said that trust is "the willingness of a party to be vulnerable to the actions of another party based on the expectation that the other will perform a particular action important to the trustor, irrespective of the ability to monitor or control that other party".

It is therefore important to ask: what kind of trust do consumers have? According to some previous studies [19,20], trust is largely divided into credibility and benevolence depending on the nature of the message conveyed by the brand. Ref. [21] defined benevolence as a degree of belief that the trustee will perform an action that will be beneficial to the exchange counterpart, irrespective of the trustee's direct interests. It can be seen as a belief based on the honesty of the exchange counterpart in fulfilling the promised obligation. On the other hand, Ref. [18] defined credibility as an objective belief based on the expertise—which is required to perform the task effectively—of the exchange counterpart, which means that professional expertise is required as a premise of the factors that lead to credibility.

These arguments are fairly generalized concepts in understanding trust. It was determined that it was reasonable to distinguish between reliability-based expertise, such as the brand's technological power, capability, and ability, and reliability-based benevolence, which is the belief that the brand intends to benefit the other party, rather than focusing on a

self-centered benefit motive. However, it should also be taken into account if the economic, social, and environmental messages of the sustainable strategy intended to be conveyed through single person media are delivered in a way with which consumers can easily sympathize for instance, simple social responsibilities [22,23]. Based on this theoretical background, we have established the following hypotheses:

**H1-1.** *The economic responsibility strategy of the sports apparel brand conveyed through single-person media will have a positive (+) impact on the expertise reliability.*

**H1-2.** *The economic responsibility strategy of the sports apparel brand conveyed through single-person media will have a positive (+) impact on the benevolence reliability.*

**H1-3.** *The environmental responsibility strategy of the sports apparel brand conveyed through single-person media will have a positive (+) impact on the expertise reliability.*

**H1-4.** *The environmental responsibility strategy of the sports apparel brand conveyed through single-person media will have a positive (+) impact on the benevolence reliability.*

**H1-5.** *The social responsibility strategy of the sports apparel brand conveyed through single-person media will have a positive (+) impact on the expertise reliability.*

**H1-6.** *The social responsibility strategy of the sports apparel brand conveyed through single-person media will have a positive (+) impact on the benevolence reliability.*

### 2.2. Brand Reliability and Purchase Intention

Purchase intention can be defined as the consumer's willingness to purchase a product when it is provided for commercial sale [17]. The purchase intention of consumers is a typical indicator used to predict purchase behavior, and brands have been continuously studying it to determine the effectiveness of marketing activities. According to a study by [24], brand reliability has a positive impact on the purchase intention of consumers, and consumers show more favorable attitudes toward advertisements of brands with high reliability than those with low reliability. In another study that investigated the relationship between reliability and purchase intention [25,26], corroborated that reliability has a positive effect on product purchase intention. This means that if consumers trust the information provider, they will consider purchasing their products.

In theory, there are a variety of major determinants that affect the purchase intention of the product that the brand promotes to consumers, but trust in the brand or product is defined as the main factor [27,28]. This is because if there is expertise reliability within a certain brand or product, or a favorable consumer belief based on experiences and knowledge, the purchase intention of consumers occurs even if it does not actually lead to purchasing. In the long run, this may lead to the purchasing behavior or affect other people's purchase intentions.

However, this path's demonstrability depends on the medium conveying the sustainable strategy to consumers [28,29]. In particular, when introducing a product that requires a significant amount of explanation, such as one involving a sustainable strategy, the role of single-person media is very large, as it can deliver the information about the sustainable strategy more intensively and effectively than offline advertisements, which are seen only for a short duration on the streets. This is because the amount and quality of information that consumers encounter online and remember are higher than those encountered offline. In fact, an experiment comparing websites where Internet users can exchange their opinions about products and the brand's websites, [29] showed that the information obtained from websites where users can freely communicate is more reliable and has a large impact on the purchase intention.

Single-person media has the advantage of allowing one to empirically check the viewers' trust in the channels that they regularly watch based on their preferences and direct interactions with the content [30,31]. For example, if a viewer of single-person media has trust in a particular channel, he/she uses a function called "subscribe" to save the channel onto their list of preferred channels, thus ensuring that the channel's content will recurrently appear. If information about the products of a sports apparel brand that uses sustainable management strategies is conveyed through a single-person media channel with many subscribers, it can be predicted that a path will form where the purchase intention of the channel subscribers may manifest [28,29]. Furthermore, if trust is solidified in the consumers towards the sports apparel products through these single-person media channels, their purchase intention could even be expected. Based on this theoretical background, we have established the following research hypotheses:

**H2-1.** *The expertise reliability of the sports apparel brand formed through single-person media will have a positive (+) impact on the purchase intention of viewers.*

**H2-2.** *The benevolence reliability of the sports apparel brand formed through single-person media will have a positive (+) impact on the purchase intention of viewers.*

The research model, includes H1-1, H1-2, H1-3, H1-4, H1-5, H1-6, H2-1, H2-2, is shown as Figure 1.

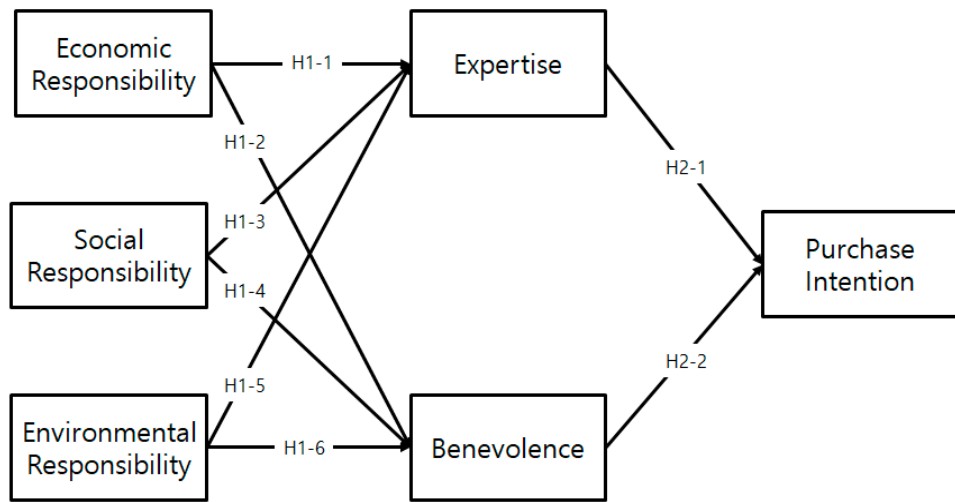

**Figure 1.** Research model.

## 3. Research Methods

### 3.1. Research Subjects

This study was conducted by a five-member research team composed of one researcher and four assistant researchers to achieve its goals. The target population of this study was four.

Twenty students who received online sports practice classes in both the semesters in 2020. An online questionnaire survey was conducted to avoid the risk of COVID-19 infection. The questionnaire was compiled by generating links using Google Survey. The questionnaire with the generated links was distributed via social media such as WhatsApp and KakaoTalk applications, and emails were retrieved. The questionnaire was retrieved from 292 respondents. Out of these, 272 questionnaires were used for analysis after excluding 20 responses due to insincere, such as returning in less than 5 min with fully answered or incomplete responses (Table 1).

**Table 1.** Results of Demographic Analysis.

| Contents | | # of Cases | % |
|---|---|---|---|
| Gender | Male | 149 | 54.8 |
| | Female | 123 | 40.2 |
| Ages | Under 19 | 63 | 23.2 |
| | 20~29 | 78 | 28.7 |
| | 30~39 | 62 | 22.8 |
| | 40~49 | 36 | 13.2 |
| | Higher than 50 | 33 | 12.1 |
| How long you watch 1 person media usually a day | Less than 10 min | 33 | 12.1 |
| | 10 min~30 min | 53 | 19.5 |
| | 30 min~1 h | 85 | 31.3 |
| | 1 h~2 h | 57 | 21.0 |
| | More than 2 h | 44 | 16.2 |
| The Routes you take the information about sports apparel | Cellphone (Tablet PC) | 102 | 37.5 |
| | Personal Computer (Laptop) | 72 | 26.5 |
| | Traditional media (TV, Radio) | 16 | 5.9 |
| | Surroundings(relatives) | 73 | 26.8 |
| | Etc. | 9 | 3.3 |
| Total | | 272 | 100.0 |

### 3.2. Usage of Stimulus

The selection process of an adequate stimulus for a successful survey in this study was a very important procedure. Instead of creating a new video stimulus, we chose an existing promotional video of a sports apparel brand and used it in the survey.

This was because the content of the existing promotional video would most likely provide a sense of familiarity to the research subjects, and therefore not create unnecessary obstacles in answering the questionnaire. The link was sent through a text message or application for convenience, after having obtained the consent in advance for watching the video for the survey. The research subjects were asked to answer the questionnaire after finishing watching the linked video. The link to the stimulus used in this study is as follows: https://www.youtube.com/watch?v=Gr0_6Vli1xE (accessed on 6 June 2022).

### 3.3. Research Tools

A questionnaire was used as the survey method, and the questionnaire scale used is summarized (Table 2).

**Table 2.** Results of Demographic Analysis.

| Variables | Sub-Variables | # of Questions | Resource |
|---|---|---|---|
| Sustainable Management | Economic Responsibility | 4 | |
| | Social Responsibility | 4 | [31] |
| | Environmental Responsibility | 6 | |
| Brand Reliability | Expertise | 3 | |
| | Benevolence | 3 | [32] |
| Purchase Intention | | 4 | [33] |

### 3.4. Analysis Procedure

The collected data were analyzed using SPSS 230.0 and AMOS 210.0. SPSS 230.0 was used for frequency analysis to analyze demographic characteristics, and reliability analysis and correlation analysis were performed to test the reliability and multi-collinearity of the questionnaire, respectively. AMOS 210.0 was used for confirmatory factor analysis

(CFA) to establish the validity of the latent variables to test the validity between factors, the causal relationships between variables, and the hypotheses. After checking the parameter estimates and GOF (Goodness of Fit) of the measurement model based on the values of CFI (comparative fit index), TLI (Tucker-Lewis Index), and RMSEA (root mean square error of approximation), the validity between the observation and latent variables was tested. Lastly, hypotheses were accepted or refused by testing the causal relationships between the variables for the structural modeling analysis.

### 3.5. The Validity and Reliability of the Measured Variables

The CFA of each research unit revealed that item 4 of the variable "Economic Responsibility" failed to explain the variance of their respective factors. The questionnaire was constructed for the use of this study based on the data provided in previous research. After the questionnaire was prepared for data analysis, its content validity was established by two professors of sports education and a research methodologist holding a Ph.D. in sports administration and marketing. In addition, a CFA was performed to test the convergent and discriminant validity, and the internal consistency of each item was evaluated by Cronbach's α coefficient. The CFA was performed primarily on each latent variable and then applied to the entire model according to the results of the CFA of the latent variables. The CFA and reliability test results are outlined in Tables 3 and 4.

**Table 3.** Results of Confirmative Factor Analysis I and Reliability Analysis.

| Variables | Contents (This Is....) | Est | SE | CR | AVE | α |
|---|---|---|---|---|---|---|
| Economic Responsibility | 1. A brand that continuously performs quality control. | 0.876 | 0.287 | | | |
| | 2. A brand that contributes to national economic growth. | 0.875 | 0.250 | 0.908 | 0.766 | 0.913 |
| | 3. A brand that strives to create jobs. | 0.875 | 0.267 | | | |
| Social Responsibility | 5. A brand that cooperates with local communities, schools, or institutions. | 0.864 | 0.341 | | | |
| | 6. A brand that supports sports and cultural activities. | 0.880 | 0.306 | 0.918 | 0.738 | 0.918 |
| | 7. A brand that engages in a lot of volunteer work for communities. | 0.856 | 0.360 | | | |
| | 8. A brand that gives back to society for the sake of humanity. | 0.835 | 0.385 | | | |
| Environmental Responsibility | 9. A brand that engages in a lot of environmental protection activities. | 0.762 | 0.564 | | | |
| | 10. A brand that produces a lot of environmentally friendly products. | 0.863 | 0.247 | | | |
| | 11. A brand that engages in a lot of environmental protection campaigns. | 0.851 | 0.281 | 0.938 | 0.715 | 0.935 |
| | 12. A brand that sponsors environmental projects. | 0.877 | 0.254 | | | |
| | 13. A brand that uses environmental resources effectively. | 0.851 | 0.294 | | | |
| | 14. A brand that makes efforts to conserve the environment | 0.866 | 0.277 | | | |
| Expertise | 15. A brand that has unrivaled technology. | 0.818 | 0.457 | | | |
| | 16. A brand that has very superior expertise. | 0.796 | 0.529 | 0.856 | 0.664 | 0.854 |
| | 17. A brand that has know-how in the sports apparel industry. | 0.831 | 0.387 | | | |
| Benevolence | 18. A brand that delivers truth to consumers. | 0.876 | 0.300 | | | |
| | 19. I think this is an honest brand. | 0.882 | 0.283 | 0.911 | 0.733 | 0.910 |
| | 20. A brand is more ethical than other apparel brands. | 0.879 | 0.315 | | | |
| Purchase to intention | 21. I would like to recommend others to buy from this brand. | 0.740 | 0.430 | | | |
| | 22. Even if there are similar products, I want to purchase this brand's product. | 0.756 | 0.448 | 0.845 | 0.578 | 0.844 |
| | 23. Even if the price is a little high, I want to purchase this brand's product. | 0.804 | 0.378 | | | |
| | 24. Sustainability of the apparel brand is important when it comes to purchasing. | 0.738 | 0.517 | | | |
| $\chi^2$ = 3580.911, df = 215, TLI = 0.964 CFI = 0.969. RMSEA = 0.050 | | | | | | |

**Table 4.** Results of Confirmative Factor Analysis II.

| Variables | | Early Qs | Final Qs | $\chi^2$ | *Df* | TLI | CFI | RMSEA |
|---|---|---|---|---|---|---|---|---|
| Sustainable Management | Economic | 4 | 3 | | | | | |
| | Social | 4 | 4 | 165.1 | 74 | 0.964 | 0.970 | 0.067 |
| | Environment | 6 | 6 | | | | | |
| Reliability | Confidence | 3 | 3 | 18.1 | 8 | 0.982 | 0.991 | 0.068 |
| | Self-Control | 3 | 3 | | | | | |
| Purchase Intention | | 4 | 4 | 3.4 | 2 | 0.990 | 0.997 | 0.050 |
| Total | | 24 | 23 | 358.911 | 215 | 0.969 | 0.964 | 0.050 |

## 4. Results

### 4.1. Results of Correlation Analysis

The Pearson correlation analysis was performed to evaluate the correlations between the quality of the class, academic self-efficacy, and class satisfaction perceived by students who took online sports practice classes. The correlations between these variables were partially significant and no multi-collinearity was observed between the variables, with the correlation coefficients not exceeding 0.80 (Table 5).

**Table 5.** Results of Correlation Analysis.

| | Economic | Social | Environmental | Expertise | Benevolence |
|---|---|---|---|---|---|
| Social | 0.676 ** | | | | |
| Environmental | 0.195 ** | 0.156 ** | | | |
| Expertise | 0.316 ** | 0.296 ** | 0.672 ** | | |
| Benevolence | 0.380 ** | 0.338 ** | 0.561 ** | 0.651 ** | |
| Purchase | 0.397 ** | 0.452 ** | 0.118 ** | 0.160 ** | 0.323 ** |

** $p < 0.01$.

### 4.2. Fit of Entire Model

For model fit indices, the maximum likelihood (ML) parameter estimation method was used. With all values exceeding the cut-off values, the adequacy of the research model was established (Table 6).

**Table 6.** Fit of Entire Model.

| Model # | $\chi^2$ | *df* | TLI | CFI | RMSEA |
|---|---|---|---|---|---|
| 1 | 440.731 | 219 | 0.945 | 0.952 | 0.059 |

### 4.3. Results of Structural Modeling Analysis

Among all the hypotheses formulated in this study, the hypothesis test results for those regarding direct effects are as follows: (1) H1-1 on the effect of the economic responsibility perceived by single person media viewer on their perceived expertise reliability was denied statistically ($t = 0.285$); (2) H1-2 on the effect of the social responsibility perceived by single person media viewer on their perceived expertise reliability was accepted statistically ($t = 2.737$, $p < 0.05$); (3) H1-3 on the effect of the environmental responsibility perceived by single person media viewer on their perceived expertise reliability was accepted statistically ($t = 11.322$, $p < 0.001$); (4) H1-4 on the effect of the environmental responsibility perceived by single person media viewer on their perceived benevolence reliability was accepted statistically ($t = 2.232$, $p < 0.02$); (5) H1-5 on the effect of the social responsibility perceived by single person media viewer on their perceived expertise reliability was accepted statistically ($t = 2.274$, $p < 0.05$); (6) H1-6 on the effect of the environment responsibility perceived by single person media viewer on their perceived expertise reliability was denied statistically ($t = 9.782$, $p < 0.001$); (7) H2-1 on the effect of the expertise reliability perceived by

single person media viewer on their perceived purchase intention was denied statistically ($t = -1.499$); (8) H2-2 on the effect of the benevolence reliability perceived by single person media viewer on their perceived purchase intention was accepted statistically ($t = 5.372$, $p < 0.001$) (Table 7).

**Table 7.** Results of Analysis on direct Effects.

| H | Path | | | Estimates | SE | T | Results |
|---|---|---|---|---|---|---|---|
| 1-1 | Economic | → | Expertise | 0.022 | 0.077 | 0.285 | Rejected |
| 1-2 | Social | → | Expertise | 0.210 | 0.077 | 2.737 ** | Accepted |
| 1-3 | Environmental | → | Expertise | 0.737 | 0.065 | 11.322 *** | Accepted |
| 1-4 | Economic | → | Benevolence | 0.194 | 0.087 | 2.232 * | Accepted |
| 1-5 | Social | → | Benevolence | 0.195 | 0.086 | 2.274 * | Accepted |
| 1-6 | Environmental | → | Benevolence | 0.632 | 0.065 | 9.782 *** | Accepted |
| 2-1 | Expertise | → | Purchase | −0.094 | 0.063 | −1.499 | Rejected |
| 2-2 | Benevolence | → | Purchase | 0.321 | 0.060 | 5.372 *** | Accepted |

* $p < 0.05$, ** $p < 0.01$, *** $p < 0.001$.

## 5. Discussions

This study aimed to empirically verify how the sustainable management strategies of a sports apparel brand promoted through single-person media affects the brand reliability perceived by consumers and their purchase intention. The results of this study are as follows:

First, we examined the influential relationships between the sustainable management strategy of the sports apparel brand promoted through single person media and the brand reliability perceived by consumers and their purchase intention, which yielded the following results: Among the brand's sustainable management activities, economic responsibility did not have a significant effect on the expertise credibility perceived by consumers (0.022), but social (0.210) and environmental responsibilities (0.737) had statistically significant positive (+) effects on the perception of expertise reliability. However, the economic (0.194), social (0.195), and environmental (0.632) responsibilities all had statistically significant positive (+) impacts on the benevolence reliability.

This result can be summarized in two main points. Firstly, while the sports apparel brand's sustainable economic strategies as a brand's expertise had little bearing on single-person media viewers' perception, the viewers valued social and environmental responsibility. Putting it more simply, the viewers accepted the social and environmental responsibility as an aspect of creditable expertise sustainability. Since sports apparel brands pursue profits as their end goal, their business activities are often directly linked to achieving this objective [34,35]. This seems to be shared among consumers, given that our results that demonstrated no significant effect of economic responsibility on perceived expertise reliability. It seems that the viewers do not favor the expression of sports apparel brands' pursuit of profits [36–38]. Thus, in terms of the public promotion of a brand's sustainable strategies, the stand out expressions of economic purposes of the sports apparel brand should be minimized as a professional corporation for contributing to the development of society [39]. Secondly, the viewers showed favorable attitudes toward all activities surrounding economic, social, and environmental responsibilities. It was demonstrated that, in principle, when a sports apparel brand performed effective management, extending beyond the level of profiting through sales of goods and services, the company could gain empathy from the viewers as it fulfills its responsibilities and obligations as a member of society [40]. Thus, in terms of the public promotion of a brand's sustainable strategies, the comprehensive, not focused on one, activities of sustainable purposes of the sports apparel brand would be favored by the viewers as a professional corporation for Contributing to the development of society [40,41]. However, as it has to contain a lot of content, how to express it will be an important issue.

Secondly, the consumers' perceived expertise reliability ($-0.094$) of the sports apparel brand that promoted their sustainable management strategies through single-person media did not have a statistically significant impact on the purchase intention of single-person media viewers. On the other hand, the consumers' perceived benevolence reliability of these brands (0.321) had a statistically significant impact on the purchase intention of single-person media viewers. These research and analysis results imply that single-person media viewers make very realistic judgments. Even if viewers prefer comprehensive sustainable management strategies of a company, including those surrounding economic, social, and environmental responsibilities but unless the expertise reliability and the benefits that the viewers can expect are adequately instilled, the viewers would not purchase [42].

Furthermore, considering that most products of sports apparel brands using sustainable management strategies, such as Patagonia, Freitag, and even other sports brands like Nike or Adidas, are in a high price range, it is possible to make a phenomenological interpretation that the viewers would have the purchase intention only when in situations in benefits, like purchasing reputable brand cheaper than others, is expected [43,44]. These results show that sports apparel's sustainable management strategies are effective and have been supported partially by some previous studies [44–46].

Therefore, we propose the following suggestion for sports apparel brands that want to convey their sustainable management strategies through single-person media [47,48]: Firstly, it can be said that the single-person media-based promotion strategy of the sports apparel brand is quite efficacious. Therefore, in the case of a new brand or brand with a low awareness, significantly high effectiveness in increasing awareness can be expected if a company sponsoring single-person media to carry out promotional activities [49].

Secondly, when it comes to the sustainable strategies of sports apparel brands using single-person media, the target viewers of single-person media should be taken into account in the pricing strategy [50,51]. The main viewers of single-person media should be identified, as well as their viewing purposes and consumption propensities while conducting market research. The basis for the promotional content of the sustainable management strategies should be created based on this.

## 6. Conclusions

Around the world, sustainable management strategies that have strayed from traditional strategies, in that they emphasize social responsibilities along with the brand's profit maximization, have become indispensable for brands in their pursuit of profits. Brands are striving to pursue more profits through sustainable management strategies, which can be used to easily win the favor of consumers, especially for sports apparel brands that consumers repeatedly purchase and use. With the prevalence of smartphones and tablet devices, single-person media has grown into a very effective networking tool in all fields around the world. In some cases, single-person media shows a higher public promotion effect than the world's leading sports broadcasting companies, such as NBC and ESPN, and has grown into an indispensable means of public promotion across the industry. Moreover, the COVID-19 pandemic acted as an environmental change that strengthened the influential power of single person media.

Academic research is the systematic formulation of an author's argument on a particular topic in a logically coherent form, the results of which must be appropriately applied in the real world. A sports apparel brand's sustainable management strategy through one-person media can be a very effective application method that maximizes its feasibility. In particular, single-person media is special because it does not require high costs and can expect immediate effects. Based on the background of these environmental changes, we found that it was necessary to verify the effectiveness of the efforts made by brands to publicly promote their sustainable management strategies through single-person media. This study, therefore, seeks to establish a theoretical framework and pathway by which these strategies establish perceived reliability among consumers. It is expected that the

findings of this study will serve as an important resource for the methods of utilizing sustainable management strategies among sports apparel brands in the future.

In the course of this study, we have found several limitations, which are summarized as follows

Firstly, this study was conducted targeting single-person media viewers after the outbreak of COVID-19 to validate the effectiveness of the sustainable management strategies conveyed through single-person media. Therefore, the direct application of this study to the sustainable management strategies of typical sports apparel bands may be subject to considerable errors as the situation surrounding the pandemic changes. In follow-up studies, it may be important to compare the results of this study to those found after COVID-19.

Secondly, this study was conducted on sports apparel brands. Therefore, the feasibility of applying its results to apparel brands in other areas of fashion is not high. For the application of non-face-to-face practical studies in other areas, the survey questions should be modified, and a proper understanding of sustainable management strategies in other apparel businesses is required. Above all, the special nature of the pertinent field of sports apparel, such as low regular use frequency or high necessity of special materials, must be well-reflected. In a follow-up study, the relationship between sustainable management strategies and apparel brands should be standardized, and big data research should be conducted to theorize it.

**Author Contributions:** Conceptualization, K.-Y.L. and T.C. methodology, T.C. and U.K.; software, T.C.; validation, formal analysis, T.C. and U.K.; investigation, T.C.; resources, U.K.; data curation, T.C. and U.K.; writing—original draft preparation, K.-Y.L. and T.C.; writing—review and editing, K.-Y.L., T.C. and U.K.; project administration, T.C.; funding acquisition, K.-Y.L. and T.C. All authors have read and agreed to the published version of the manuscript.

**Funding:** This research received no external funding.

**Institutional Review Board Statement:** Not applicable.

**Informed Consent Statement:** This research received Informed Consent Statement from the all research subjects.

**Data Availability Statement:** Not applicable.

**Conflicts of Interest:** The authors declare no conflict of interests.

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
