# Peer review of "The Impact of Sustainable Management Strategies of Sports Apparel Brands on Brand Reliability and Purchase Intention through Single Person Media during COVID-19 Pandemic: A Path Analysis"

_sustainability, doi:10.3390/su14127076_

Round 1

Reviewer 1 Report

[background]
Connecting is needed to how single-person media can help sustainable management. Or maybe there's a specific explanation.

It seems necessary to explain how a single-person media communicates multiple responses.

[Methods]
In Results of Demographic Analysis, the total amount of Ages does not seem to fit.

The researcher conducted an experimental study using stimulants, but the video is 22 minutes long, which seems too long for all 300 participants to see.

I recommend you edit the video to produce an appropriate length of stimulation.

Author Response

1. In this study, single person media is described as a new information delivery tool and media type that delivers information about a company's enterprising image and role to consumers and induces consumption. This is because the viewers can watch at anytime they want unlike conventional TV or radio, and the information they receive through the single person media with unrefined expressions could have a huge impact realistically. This part is written on the second page, and it has been modified by adding a sentence 

2. An explanation has been added to the last paragraph of the introduction

3. The figures for age classification in Table 1 have been corrected.

4. You are right. However, in consideration of the COVID-19 situation, the survey was conducted online survey, allowing respondents to freely view and respond online.

5. In future research, we will reflect the opinions of the judges in the method of producing stimulants for sustainable management.

Reviewer 2 Report

Moderate English changes are required.

 The introduction section should be improved.

 The discussion section should be written clearly. Mention the key findings.

 Mention what the Academic implications of the paper are.

 Table 6: use the word "Rejected" instead of "Denied"

Author Response

1. This article has been translated through a professional translator. The revision was rechecked.

2. The point of the introduction has been checked once again.

3. The logical flow of the discussion section has been rechecked, and some parts of page 10 have been corrected.

4. Added the academic meaning of the thesis to the conclusion paragraph.

Reviewer 3 Report

This study aims to analyze the path through which the sustainable management strategy established by a sports apparel brand affects the brand awareness and reliability and the purchase intention of consumers when experiencing this strategy through single-person media.

The authors rightly noted that the results of this study will serve as an important resource for the methods of utilizing sustainable management strategies among sports apparel brands in the future.

The literature review in the field of research and the conclusions provided by the authors are interesting.

Research methods are correct and meaningful.

The article should be published.

Author Response

1 We the authors appreciate for suggesting a good evaluation for insufficient research. We pray that the judges will be filled with only good things.

2. Part of the paper has been corrected and marked in red.
